# A Comparison of Bioinformatics Pipelines for Enrichment Illumina Next Generation Sequencing Systems in Detecting SARS-CoV-2 Virus Strains

**DOI:** 10.3390/genes13081330

**Published:** 2022-07-26

**Authors:** Stefanus Bernard, Hendra Wibawa, Mohamad Saifudin Hakim, Arli Aditya Parikesit, Chandra Kusuma Dewa, Yasubumi Sakakibara

**Affiliations:** 1Department of Computer Science and Electronics, Faculty of Mathematics and Natural Sciences, Universitas Gadjah Mada, Yogyakarta 55281, Indonesia; 2Department of Bioinformatics, School of Life Sciences, Indonesia International Institute for Life Sciences, Jakarta 13210, Indonesia; stefanus.bernard@alumni.i3l.ac.id (S.B.); arli.parikesit@i3l.ac.id (A.A.P.); 3Pediatric Surgery Division, Department of Surgery or Genetics Working Group or Translational Research Unit, Faculty of Medicine, Public Health and Nursing, Universitas Gadjah Mada, Yogyakarta 55281, Indonesia; drgunadi@ugm.ac.id; 4Indonesian Young Academy of Science (ALMI), Jakarta 10110, Indonesia; 5Disease Investigation Center, Wates, Yogyakarta 55602, Indonesia; hendra.wibawa@pertanian.go.id; 6Department of Microbiology, Faculty of Medicine, Public Health and Nursing, Universitas Gadjah Mada, Yogyakarta 55281, Indonesia; m.s.hakim@ugm.ac.id; 7Genetics Working Group, Faculty of Medicine, Public Health and Nursing, Universitas Gadjah Mada, Yogyakarta 55281, Indonesia; marcellus@mail.ugm.ac.id; 8Department of Informatics, Faculty of Industrial Technology, Universitas Islam Indonesia, Yogyakarta 55584, Indonesia; chandra.kusuma@uii.ac.id; 9Department of Biosciences and Informatics, Keio University, Yokohama 223-8522, Japan; yasu@bio.keio.ac.jp

**Keywords:** SARS-CoV-2, Next Generation Sequencing, enrichment, Illumina, bioinformatics pipeline

## Abstract

Severe Acute Respiratory Syndrome Coronavirus 2 (SARS-CoV-2) is a newly emerging virus well known as the major cause of the worldwide pandemic due to Coronavirus Disease 2019 (COVID-19). Major breakthroughs in the Next Generation Sequencing (NGS) field were elucidated following the first release of a full-length SARS-CoV-2 genome on the 10 January 2020, with the hope of turning the table against the worsening pandemic situation. Previous studies in respiratory virus characterization require mapping of raw sequences to the human genome in the downstream bioinformatics pipeline as part of metagenomic principles. Illumina, as the major player in the NGS arena, took action by releasing guidelines for improved enrichment kits called the Respiratory Virus Oligo Panel (RVOP) based on a hybridization capture method capable of capturing targeted respiratory viruses, including SARS-CoV-2; therefore, allowing a direct map of raw sequences data to SARS-CoV-2 genome in downstream bioinformatics pipeline. Consequently, two bioinformatics pipelines emerged with no previous studies benchmarking the pipelines. This study focuses on gaining insight and understanding of target enrichment workflow by Illumina through the utilization of different bioinformatics pipelines named as ‘Fast Pipeline’ and ‘Normal Pipeline’ to SARS-CoV-2 strains isolated from Yogyakarta and Central Java, Indonesia. Overall, both pipelines work well in the characterization of SARS-CoV-2 samples, including in the identification of major studied nucleotide substitutions and amino acid mutations. A higher number of reads mapped to the SARS-CoV-2 genome in Fast Pipeline and merely were discovered as a contributing factor in a higher number of coverage depth and identified variations (SNPs, insertion, and deletion). Fast Pipeline ultimately works well in a situation where time is a critical factor. On the other hand, Normal Pipeline would require a longer time as it mapped reads to the human genome. Certain limitations were identified in terms of pipeline algorithm, whereas it is highly recommended in future studies to design a pipeline in an integrated framework, for instance, by using NextFlow, a workflow framework to combine all scripts into one fully integrated pipeline.

## 1. Introduction

China’s authority reported patients associated with pneumonia derived from unknown etiology in Wuhan back in December 2019. It was identified as a new type of coronavirus and successfully isolated and fully sequenced on 10 January 2020, named Severe Acute Respiratory Syndrome Coronavirus 2 (SARS-CoV-2). It enters the body through receptors called Angiotensin-Converting Enzyme-2 (ACE-2), widely expressed in human organs, including lower respiratory tract organs such as lungs [1]. Following entry, the human body will trigger protective responses and eventually cause acute respiratory failure with more serious complications. This disease was eventually termed Coronavirus Disease 2019 (COVID-19). Understanding of morphological and viral genome characteristics of SARS-CoV-2 provides valuable insights to help address the worsening pandemic situation; however, transmission and anti-viral treatments might induce mutations and consequently generate more virulent strains with higher fatalities or resistance to available treatment and vaccines [2]. One study has conducted data science analysis towards SARS-CoV-2 genome submissions between February and May 2020 and revealed that several variants exist with D614G, where adenine substitution to guanine happens at position 23,403. It is the most common variant discovered since December 2019 [2]. SARS-CoV-2 variant identification is pivotal in providing insight into viral infectivity, severity, and also in studying the evolutionary analysis of SARS-CoV-2.

The COVID-19 pandemic has brought computational biology with Next Generation Sequencing (NGS) to the frontline as it revolutionized the biological sciences in the past decades with its high throughput and tremendous ability to study biological systems through a wide variety of applications. NGS enables researchers to conduct Whole Genome Sequencing (WGS), the construction of a complete DNA sequence belonging to an organism’s genome at a single time. The application of WGS is capable of understanding the transmission pattern, gaining insight into outbreak control decisions, and discovering new variants of viruses [3]. This was proven when WGS was capable of helping public health decision-making strategy during the 2014–2016 West African Ebola outbreak; therefore, WGS studies during the ongoing COVID-19 pandemic is an active area of research. The first complete genome of SARS-CoV-2 was fully recovered on 10 January 2020 through de novo assembly using metagenomic RNA sequencing [4]. Afterwards, 11,601,013 whole genome sequences of SARS-CoV-2 were submitted to Global Initiative on Sharing Avian Influenza Data (GISAID); data sharing with Indonesia reported 25,817 complete genomes of SARS-CoV-2 as of January 2021.

NGS technologies are heavily influenced by Illumina^®^ as the prominent player in second-generation NGS. All Illumina’s NGS platforms were built based on bridge amplification with ease of support and are applicable to genomic sequencing, exome sequencing, targeted sequencing, metagenomics, and RNA sequencing [5]. Responding to the COVID-19 pandemic, Illumina published a guideline as the improvement for target enrichment workflow in detecting respiratory viruses using the NGS platform. The workflows are highly sensitive and able to characterize common respiratory viruses, including coronavirus strains, without the need to map raw NGS data to the human genome [6]. Target enrichment has been widely used long before the COVID-19 pandemic; it utilizes hybrid-capture methods to capture genomic regions of interest using biotinylated oligonucleotide probes designed to hybridize regions of interest [7]. Furthermore, its sensitive detection excludes the need for high read depth required for shotgun metagenomic sequencing [8]. 

Target enrichment workflow through a hybrid-capture method and is able to directly detect respiratory viruses; however, no previous studies evaluated how accurate the target enrichment workflow guideline provided by Illumina is in detecting SARS-CoV-2. This study incorporates different bioinformatics pipelines for target enrichment workflow in detecting SARS-CoV-2 using the Illumina NGS system. The aim of this study is to compare different bioinformatics pipelines toward the target enrichment workflow by Illumina. The NGS data were obtained from eight hospitalized patients in Yogyakarta and Central Java, who tested positive for SARS-CoV-2 and took a Real Time-Polymerase Chain Reaction (RT-PCR) swab test between May and September 2020. Prior to joining the study, patients were given informed consent and the study design was approved by the Medical and Health Research Ethics Committee of the Faculty of Medicine, Public Health, and Nursing, Universitas Gadjah Mada, alongside Dr. Sardjito Hospital (KE/FK/0563/EC/2020). The first pipeline, dubbed as ‘Fast Pipeline’, directly maps the raw NGS data to the SARS-CoV-2 reference genome. The second ‘Normal Pipeline’ maps the raw NGS data to the human genome and proceeds to map subsequent unmapped reads to the SARS-CoV-2 reference genome. The comparison between pipelines observed the identification of nucleotide substitutions and amino acid mutations.

## 2. Materials and Methods

### 2.1. Viral RNA Extractions and Library Preparation for Whole Genome Sequencing

Viral sampling, library preparation, and WGS were fully performed by the Genetics Working Group (Pokja Genetik) of the Faculty of Medicine, Public Health, and Nursing, Universitas Gadjah Mada, alongside the Disease Investigation Center, Wates, Yogyakarta. Virus samples in this study were collected through nasopharyngeal swabs of hospitalized patients with COVID-19 between May to December 2020 in Yogyakarta and Central Java provinces, Indonesia.

Viral samples were placed into viral transport media immediately after being collected and sent to the Department of Microbiology, Faculty of Medicine, Public Health and Nursing, Universitas Gadjah Mada, alongside Disease Investigation Center, Wates, Yogyakarta. Viral RNA extractions and library preparation for WGS described below are in accordance with the previously published research by Gunadi et al. [9]. First, total viral RNA was extracted from nasopharyngeal swabs samples using QiAMP Viral RNA mini kit and continued by double-stranded cDNA synthesis using Maxima H Minus Double-Stranded cDNA Synthesis. This was followed by purification using the GeneJET PCR Purification kit. Library for WGS was prepared using the Nextera DNA Flex for Enrichment using the Respiratory Virus Oligos Panel. Afterward, WGS was conducted in the Illumina MiSeq instrument with MiSeq reagents v3 150 cycles. WGS results in paired-end reads of FASTQ files that were used for further bioinformatics downstream analysis processes [9].

### 2.2. Patients’ Whole Genome Sequencing Data

NGS data generated from Illumina MiSeq instruments were sent to the Department of Computer Sciences and Electronics, Faculty of Mathematics, and Natural Sciences, Universitas Gadjah Mada for downstream bioinformatics analysis. Table 1 describes the NGS data of hospitalized patients with COVID-19 that were involved in this study. 

A total of 16 patients with COVID-19 were involved in this study ranging between the age of 30 to 88. Sampling was conducted between 16 May 2020 and 27 December 2020 and divided into batch 1 (4 samples), batch 2 (4 samples), and batch 3 (8 samples), generating 16 NGS data, respectively, as observed in Table 1. In total, 8 of 16 patients have at least one comorbidity.

### 2.3. Bioinformatics Pipeline for SARS-CoV-2 Nucleotide and Amino Acids Variant Analysis

The whole bioinformatics pipeline in this study was adopted from the combination of the SARS-CoV-2 nucleotide variant analysis tutorial in Galaxy by Beek et al. [10] and the utilization of bioinformatics tools for amino acids variant analysis. Subsequently, there is one main pipeline with two branches, whereas the first branch pipeline, dubbed ‘nucleotide substitution’ used to identify nucleotide variation and the second branch pipeline, dubbed ‘amino acids substitution’ used to identify amino acids variation. 

This study would benchmark different bioinformatics NGS pipelines called ‘Fast Pipeline’, represented by Figure 1, and ‘Normal Pipeline’, represented by Figure 2. Overall, the differences between pipelines occurred in the read mapping to reference genome phase due to the nature of enrichment sequencing workflows provided by Illumina, which are able to sensitively detect respiratory viruses, including SARS-CoV-2 [6]. Consequently, Normal Pipeline was constructed under the assumption that applied enrichment sequencing workflows by Illumina failed to directly detect SARS-CoV-2 during WGS in the Illumina NGS platform; therefore, it uses both human genome (accession number: GRCh38) and SARS-CoV-2 (accession number: NC_045512.2) as the reference genomes, where the read mapping was conducted twice, first with the human genome and followed by attaining the unmapped reads and followed by a second read mapping with the SARS-CoV-2 genome. Fast Pipeline, by default, does not require twice read mapping as the counterpart of Normal Pipeline; we assumed the enrichment sequencing workflows successfully detected SARS-CoV-2 during WGS in the Illumina NGS platform. 

NGS paired-end data were subjected to quality control by using FASTQC and Trimmomatic as part of the NGS quality control procedure. This was followed by a mapping of paired-end reads to the SARS-CoV-2 genome using the BWA-MEM algorithm. SAM files generated from the read mapping were converted to BAM using SAMtools. Afterwards, the BAM file generated was used as the foundation for both ‘Nucleotide Substitution’ and ‘Amino Acids Substitution’ pipelines. The ensuing process would discover SNPs and amino acid mutations and it will be fully compared with subsequent mutations discovered in the Normal Pipeline.

Immediately after data acquisition, quality control of NGS paired-end data was conducted by using FASTQC and Trimmomatic. Afterwards, the paired-end reads were mapped to the human genome using the BWA-MEM algorithm. Unmapped reads were obtained by using the SAMtools and were converted back to FASTQ format using Bedtools, a flexible suite for genomic analysis [11]. Acquired reads in the form of FASTQ were subjected to second read mapping to the genome of SARS-CoV-2. Subsequent SAM files generated from second read mapping were converted to BAM using SAMtools and continued by the implementation of the ‘Nucleotide Substitution’ and ‘Amino Acids Substitution’ pipeline as previously explained above. Discovered SNPs, as well as amino acid mutation, were all compared with Fast Pipeline in order to observe whether differences in pipeline application would affect the discovered SNPs and mutated amino acids.

## 3. Results

### 3.1. Overview of Whole Genome Sequencing Data

All raw WGS data in the form of FASTQ were provided by the Intelligent Systems Laboratory under the Department of Computer Sciences and Electronics of the Universitas Gadjah Mada. Following data retrieval, all WGS data were subjected to standard quality control checking by using FASTQC. Table 2 shows the overview of WGS data involved in the study. All WGS data are paired-end reads, meaning the total sequences indicated below represent the total of both sequences from 5′ to 3′ and vice versa. Furthermore, each sample has a varying level of Cycle Threshold (CT) value from the lowest of 13.27 to the highest of 27.92 and % GC between 38–50; however, the sequence length is remarkably the same in all samples, with a range between 35–74. Total viral RNA was isolated by using the QiAMP Viral RNA mini kit (Qiagen, Hilden, Germany). The presence of SARS-CoV-2 was detected using the Real-Q 2019-nCoV Detection Kit (BioSewoom, Seoul, Korea), targeting the RdRp and E genes of SARS-CoV-2 with LightCycler 480 Instrument II (Roche Diagnostics, Mannheim, Germany). The cut-off Ct values were ≤38 for both genes.

All samples were trimmed by using Trimmomatic following quality control check in FASTQC. Default parameters were used in the trimming of bad reads. The adapter was trimmed using the default Illumina paired-end adapter reads TruSeq3-PE-3.fa. Trimmomatic, by default, will output the number of sequences that pass filtering and those discarded either because of not passing the filtering parameters or due to only one strand surviving filtering. On average, samples retained their original sequences in the form of forward and reverse pairs, with 98.3% of them passing the trimming, and the rest 1.7% were trimmed and discarded, as represented in Table 3. 

### 3.2. Comparison of Reads Distribution in Normal Pipeline and Fast Pipeline

BWA-MEM running default parameters were used to map all samples to reference genomes in Fast Pipelines. Prior to mapping, indexing was conducted to the SARS-CoV-2 genome in Fast Pipeline. Read mapping generates a SAM file format as an input file to measure the distribution of reads. These measurements were taken by using SAMtools and applicable by running SAMtools alignment statistics code after read mapping to the reference genome. First, the SAM file generated from the ensuing mapping process was converted to a BAM file. This was continued by sorting the BAM file and running the SAMtools alignment statistics code. Alignment statistics generated by SAMtools were appended to text files, compiled, and summarized—shown in Table 4. As previously mentioned, Fast Pipeline directly maps the samples to the SARS-CoV-2 genome (accession number: NC_045512.2). Consequently, the distribution of reads was divided only into two categories—those that were fully mapped to the SARS-CoV-2 genome and those that were not. In total, 9 out of 16 samples with the NGS code of B6, C5 from Batch 1, S3 from Batch 2, and S3-1, S3-4, S3-5, S3-7, S3-8, S3-11 from Batch 3 have at least >50% of reads fully mapped to SARS-CoV-2 genome; the rest of the samples pose <50% of reads fully mapped to the SARS-CoV-2 genome. 

Normal Pipeline read mapping utilizes the same BWA-MEM tools and was run using the default parameters. All samples were mapped to the human genome (accession number: GRCh38); unmapped reads were acquired and mapped immediately to the SARS-CoV-2 genome (accession number: NC_045512.2). Both the human genome and SARS-CoV-2 genome were indexed as the basis for read mapping prior to alignment. Furthermore, the number of reads were counted during the all-read mapping procedure as well as during BAM conversion back to FASTQ prior to the second round of read mapping. The distribution of reads resulting from the Normal Pipeline bears a resemblance to the Fast Pipeline, as shown in Table 4 and Table 5. Interestingly, the proportion of unmapped reads in Fast Pipeline was, in fact, human genomes in the Normal Pipeline, as shown in Table 5. Furthermore, on average, 0.78% of total reads in all samples were derived from unknown organisms as they were neither mapped to the human genome nor to the SARS-CoV-2 genome. In addition, other significantly low reads were skipped during BAM conversion back to FASTQ due to unidentified mate pairs.

### 3.3. Comparison of Coverage Depth in Normal Pipeline and Fast Pipeline

Proceeding the count of reads, coverage depth is another pinpoint factor for comparison, defined as the average times that certain reads are mapped into specific regions inside full genome sequences. Coverage depth in this study represents how many occurrences the reads in samples were mapped to a specific region in the SARS-CoV-2 genome. It may be performed by the utilization of coverage statistics analysis from the ensuing SAM files generated from the read mapping, whereas it was converted into BAM files and sorted accordingly. Furthermore, it was obtained by using SAMtools by running a specific command for coverage statistics analysis. 

Table 6 represents the read mapping coverage results. Briefly, all samples own a higher number of coverage depth levels, with only Sample F2 running in Normal Pipeline having only 94.6 times coverage depth. Overall, Fast Pipeline tends to have a higher level of coverage depth in all samples if compared with normal pipelines, as observed in percentage differences results. Compellingly, the coverage depth results in all samples running both pipelines were discovered to be linear with the number of reads mapped to the SARS-CoV-2 genome, as shown in Table 4 and Table 5. 

### 3.4. Comparison of Variations Annotated Post Variant Calling

Three types of variation were annotated post-variant calling, including SNPs, insertion, and deletion. Subsequent variants derived from both pipelines were compiled, counted, and plotted in the form of a bar stack chart, as shown in Figure 3 and represented in more detail in Table 7. 

An abundant number of variations were detected in all samples running both pipelines, with only sample F2, F4, and S3-14 having significantly less variation. Furthermore, samples implemented in Fast Pipeline pose a higher number of variations if compared to Normal Pipeline, on average 15.16% for all variations, 18.11% for SNP, 11.9% for insertion, and 10.84% for deletion. These results were compared with count reads in the previous section and we interestingly discovered the number of variations in each pipeline is linear with the number of reads mapped to the SARS-CoV-2 reference genome, as represented by Table 4 and Table 5. Samples subjected to Normal Pipeline lose a considerable number of reads mapped to the SARS-CoV-2 genome as it was mapped to the human genome beforehand. As a consequence, the number of reads fully mapped to SARS-CoV-2 in Normal Pipeline is lower than in Fast Pipeline. This series of events further affect the number of successfully annotated variants where samples implemented in Fast Pipeline have more variations than their counterparts.

### 3.5. High Quality and Annotated Nucleotide Substitutions and Amino Acids Mutations

High-quality SNPs were obtained from all batch 1 samples implemented in Fast Pipeline with a threshold above 20,000 with the exception of sample F2 (batch 1) and S3-14 (batch 3) as it lacks quality SNPs above the aforementioned threshold; therefore, SNPs retrieved in sample F2 were considered if the quality threshold was above 2000; in sample S3-14, they were considered if the quality threshold was above 5000. In batch 1, the highest number was obtained from sample C5 with 17 SNPs. Others in decreasing order are sample F4 (13 SNPs), sample B6 (11 SNPs), and the lowest one is sample F2, with only two SNPs annotated. Those were mapped according to the position inside the SARS-CoV-2 genome and we discovered their presence inside four regions of 5′UTR, ORF1AB, ORF3A, ORF7A, and three glycoproteins including spike, matrix, and nucleocapsid as shown in Table 8. Most batch 2 samples own a substantial amount of high-quality SNPs if compared to all batch 1. In total, 21 SNPs were successfully annotated in sample S3 of batch 2 and considered the highest number of SNPs among all samples; others in decreasing order were sample S9 with 18 SNPs, and both sample S10 and sample S15 each with 14 SNPs, respectively. SNPs in batch 2 samples are well distributed in the SARS-CoV-2 region. Furthermore, their presence was observed inside five regions, including 5′UTR, ORF1AB, ORF3A, ORF8, ORF10, and three glycoproteins of the spike, matrix, and nucleocapsid. Table 9 represents identified SNPs in all batch 2 samples running all pipelines. Table 9a represents part 1, while Table 9b represents part 2. In batch 3, the highest number was obtained from samples S3-4, S3-5, S3-7, and S3-11 with 12 SNPs; others in decreasing order are sample S3-9 (11 SNPs), sample S3-1, S3-8 (10 SNPs), and the lowest one is sample S3-14 with only six SNPs annotated. Those were mapped according to the position inside the SARS-CoV-2 genome and we discovered their presence inside three regions of ORF1AB, ORF3A, ORF8, and two glycoproteins, including the spike and nucleocapsid, as shown in Table 10. The pink color and the bold nucleotides in the Table 8, Table 9 and Table 10 represent SNPs.

The same methodology as the Fast Pipeline was applied to all samples implemented in the Normal Pipeline. High-quality SNPs were retrieved in all samples in batch 1 and batch 2 with a quality threshold above 20,000. The same exception was applied to sample F2 as it lacks SNPs with quality above the mentioned threshold. Consequently, a quality threshold above 2000 was applied to sample F2 and a quality threshold above 5000 was applied to samples S3-14. All high-quality SNPs were collected and mapped into their respective position and region inside the SARS-CoV-2 genome. Furthermore, we conducted a comparative analysis between Fast Pipeline and Normal Pipeline in terms of successfully annotated high-quality SNPs. Interestingly, for batch 1 and batch 2, we discovered both pipelines result in identical annotated nucleotide substitution corresponding to their position and regions, as shown in Table 8 and Table 9a,b. On the other hand, for batch 3, we discovered that the normal pipeline results slightly different number of SNP than the Fast Pipeline, represented as a yellow color in Table 8. Sample S3-9 has 11 SNPs using the Fast Pipeline; it has 12 SNPs using the Normal Pipeline. Then, Sample S3-14 has six SNPs using the Fast Pipeline; it has five SNPs using the Normal Pipeline.

Overall, nucleotide substitutions in 14 out of 16 samples involved in this study have identical high-quality SNPs in both pipelines, albeit the differences in the number of variations and mapped reads as mentioned above. We carried out further analysis of amino acid substitution to compare how specific nucleotide substitution may code for different amino acids. As a process to detect the amino acid mutations, full-length genomes were constructed from each sample based on the SARS-CoV-2 reference genome (NC_045512.2). Consensus sequences were mapped to all SARS-CoV-2 regions. Table 11 shows the results of consensus sequences generated by using a combination of SAMtools and BEDtools. We discovered an interesting pattern where the consensus sequences constructed in all samples implemented in the Fast Pipeline pose full-length nucleotide lengths of 29,903 bp, the same length as those of SARS-CoV-2 reference sequences—consensus sequences representing the Normal Pipeline vary in nucleotide length. 

Table 12 represents identified amino acid mutations in batch 1 samples running all pipelines; Table 13 represents batch 2 and Table 14 represents batch 3. The pink color and the bold nucleotides in the Table 12, Table 13 and Table 14 represent amino acid mutations. Overall, batch 3 samples pose higher amino acid mutations compared to batch 1 and batch 2 samples in the Fast Pipeline. In batch 1 samples, the highest amino acid mutations were discovered in sample C5 with 10 detected mutations. Sample B6 and sample F4 pose the same five detected mutations. Sample F2 is considered to be the lowest, with only one detected mutation. These mutations are well distributed inside four regions, 5′UTR, ORF1AB, ORF3A, ORF7A, and two glycoproteins, the spike and nucleocapsid. Batch 2 samples own a significant number of mutations, with S3, S9, and S10 having the same 10 detected mutations, leaving sample S15 with only five detected mutations. They are dispersed in four regions, including 5′UTR, ORF1AB, ORF3A, ORF8, and two glycoproteins being spike and nucleocapsid. Batch 3 samples have the highest number of amino acid mutations. Sample S3-5 and S3-7 have 12 detected mutations. Then, sample S3-4 and Sample S3-11 pose 11 detected mutations, samples S3-1, S3-8, and S3-9 pose 10 detected mutations; samples S3-14 pose only six detected mutations. These mutations are well distributed inside three regions, ORF1AB, ORF3A, ORF8, and two glycoproteins, the spike and nucleocapsid. A unique finding discovered in the Fast Pipeline is an ambiguous amino acid (indicated by X) in sample C5 at position 54 inside the region of NS3-ORF3A.

Amino acid mutations detected in the Normal Pipeline resemble and are even almost identical to the Fast Pipeline. These observations and comparisons were made thoroughly to all parameters, including mutated amino acids, reference, alternate, and specific regions inside the SARS-CoV-2 genome. The only differences were six amino acid mutations observed in four samples, represented as yellow color in Table 12 and Table 14. First, in sample F2 where an ambiguous amino acid (indicated by X) was detected at position 769 inside the region of NSP12-ORF1AB; the other ambiguity in sample C5 actually reflects those in the Fast and Normal Pipeline. Second, sample S3-4 has T/Y amino acid mutation at position 386 inside the region of NSP4-ORF1AB. Third, sample S3-9 has three different amino acid mutations at position 1396 inside the region of NSP3-ORF1AB, position 43 inside the region of ORF8, and position 151 inside the nucleocapsid. Fourth, in samples S3-14, at position 57 inside the ORF3A, Q57H amino acid mutation is not detected using the Normal Pipeline. Benchmarking of runtime execution was made by calculating the time required to finish each pipeline from the beginning of quality control until the end of each branch, meaning the annotation of SNPs and detection of mutated amino acids as represented by Table 15. Custom automated bash scripts were created to count the time required for each command line in units of seconds and output it in the form of .txt files. In Python, a separate timer command also in the units of seconds was added in the Python script used. 

## 4. Discussion

Here, we present a comparison between Fast Pipeline and Normal Pipeline in terms of proportion of mapped reads and their implication towards the coverage depth and annotated variants. It showed 7 out of 16 samples (F2, F4, S9, S10, S15, S3-9, and S3-14) significantly mapped to the human genome rather than the SARS-CoV-2 genome, indicating that contamination may have occurred in the samples, as shown in Table 5. Previous studies have noted the application of NGS alongside metagenomes allows researchers to detect the presence of subjected viral pathogens; however, the direct recovery from clinical specimens such as nasopharyngeal swabs poses a great challenge owing to the possibility of contamination from the host’s genome as well as limited viral RNA quantities [12]. As a result, for countermeasures in downstream bioinformatics analysis, it is compulsory for reads mapped to the human genome to be discarded during the read mapping process, leaving the rest mapped to the respiratory virus genome. Numerous enrichment kits have been produced to separate viruses with the host genome, for example, the NetoVir and recently improved Respiratory Virus Oligo Panel by Illumina [6,13]; however, the standard indicator for respiratory virus characterization still relies on the detection of potential viral types in metagenomes [12]. This implies that the Normal Pipeline acts as the key indicator toward the Fast Pipeline, whether the results are reliable or not. 

Venturing further to count reads and coverage depth, a linear relationship was discovered between the number of reads mapped and ensuing coverage depth; it was shown that each sample bearing the Fast Pipeline tends to have higher coverage depth than its counterparts, as represented in Table 4, Table 5 and Table 6**.** The Fast Pipeline, by default, directly maps the reads towards the SARS-CoV-2 genome. As a result, most reads were retrieved intact and mapped several times, resulting in a higher coverage depth, with varying percentages from only 4.7% differences (sample B6) to the highest of 21.6% differences (sample F2) with the Normal Pipeline. On the other hand, the Normal Pipeline lost a substantial amount of reads as they were mapped to the human genome, resulting in lower coverage depth than the Fast Pipeline. Surprisingly, the mapped reads also affect the number of variants annotated, including SNPs, insertion, and deletion between pipelines. They resemble relationships as those between the number of reads and coverage depth. The Fast Pipeline has a relatively higher number of SNPs, insertion, and deletion fully annotated in all samples against the Normal Pipeline, as represented in Figure 3 and Table 7.

This study uses the same data of batch 1 samples as the previous study, and therefore, both nucleotide and amino acid substitutions identified in batch 1 samples were compared thoroughly with the previous study. Referring to all batch 1 samples and the previous study by Gunadi et al. [9], no differences either in nucleotide substitution or change in position were observed in both the Fast Pipeline and the Normal Pipeline, as shown in Table 8. All identified SNPs in the batch 1 samples are identical in terms of the number of high-quality SNPs annotated, as well as in substitution and position inside the SARS-CoV-2 region to those in Gunadi et al. [9]. 

In this study, we noticed several ambiguous amino acids following the construction of consensus sequences and translation to amino acids in the batch 1 samples. Captivatingly, these were not mentioned in the previous study and, therefore, convinced us to trace back the triplet’s codes for the ambiguity. A Triplet of nucleotide bases consisting of A, T, C, or G commonly codes for a single amino acid; however, a case where ambiguity shows up implies that there is a possibility the triplet may code for more codons [14]. The three ambiguous amino acids X discovered were as follows: each one was detected in sample C5 running both the Fast Pipeline and the Normal Pipeline (position 54; NS3-ORF3A), while another one was detected in sample F2 running the Normal Pipeline (position 769; NSP12-ORF1AB)**,** as shown in Table 12. We successfully traced back the triplet’s codes for three ambiguity bases using a Python script equipped with pandas. Furthermore, by referring to the central dogma of biology, an illustration representing it was created, as seen in Figure 4 and Figure 5.

Traced back of ambiguity at sample C5 (position 54; NS3-ORF3A) revealed a ‘GYT’ as the starting triplet codes for X amino acids (Figure 4). The International Union of Pure and Applied Chemistry (IUPAC) provided basic nomenclature for incomplete nucleotides 25 years ago, whereas the recent one has been further elucidated as an extended IUPAC code [15]. The ‘Y’ here represents pyrimidines, a heterocyclic nitrogenous base with three possible translations being C (cytosine) or T/U (thymine/uracil). Hence, two possibilities exist if ‘Y’ was changed with cytosine or thymine. The outcome of replacing ‘Y’ with cytosine would result in the translation of alanine amino acid, and therefore, it is not mutating, as it implies the same amino acid in the SARS-CoV-2 reference genome. Furthermore, it would be different if ‘Y’ was replaced with thymine as it will be translated to valine amino acid, resulting in mutated amino acids with ‘A’ (alanine) substituted to ‘V’ (valine), as shown in Figure 4. Interestingly, the previous study in sample C5 designated position 54 at region NS3-ORF3A as ‘mutated’ with the ‘V’ (valine) written in the exact position [9]. A comparison was made thoroughly, and we hypothesized the possible prominent factor, in this case, might be derived from a different read mapping algorithm. The previous study used a standard BWA-backtrack aligner embedded inside the UGENE program for batch 1 samples analysis, while we utilized BWA-MEM aligner for read mapping to all samples. BWA-backtrack aligner by default, specifically designed for Illumina sequence reads up to 100 bp with a sequencing error rate below 2%. On the other hand, BWA-MEM is the latest and most sophisticated algorithm designed for reads from 70 bp–1 Mbp equipped with more tolerated error, faster, and more accurate compared to its predecessor, the standard BWA-backtrack algorithm [16].

A similar investigation was conducted on the ambiguity discovered in sample F2 (position 769; NSP12-ORF1AB) running the Normal Pipeline. Traceback was utilized using the same Python script and method as those in sample C5. Figure 5 represents the flowchart how the possible translation result of ambiguous amino acid X detected in Sample F2 running in the Normal Pipeline. Eventually, it was revealed that the ‘NGC’ started prior to triplet translation to ambiguous amino acids. Uniquely, ‘N’ may be translated into any possible bases, either A, C, G, or T/U [15]. Consequently, there are four possible bases replacing the ‘N’, each with a different amino acid translation result. One of them bearing ‘S’ refers to serine amino acid is the same one designated in such a position within the SARS-CoV-2 reference sequence, and therefore, the aforementioned translation process will not alter the residue. Another possible three nucleotide substitutions, including T, C, and G, are codes for different amino acids, and as a result, a mutation occurs.

Surprisingly, the ambiguity occurring in sample F2 was only discovered in the Normal Pipeline. Its counterparts were coded for the same amino acids as in the previous study, where both were the same as the SARS-CoV-2 reference sequence, meaning no alteration occurred. We hypothesize this phenomenon might be due to the combination of the Normal Pipeline and different read mapping algorithms used as mentioned above.

Comparison of batch 2 samples running between the Fast Pipeline and Normal Pipeline showed no differences in either nucleotide substitution or amino acid mutations; however, we noted the difference in all samples and batches, specifically in the NSP12-ORF1AB where NSP12 inside ORF1AB poses two regions of 13,442–13,468/13,468–16,236 as mentioned in GenBank data (accession number: NC_045512.2). Differences in the region were observed between this study and the previous one by Gunadi et al. [9], where shifting in altered amino acid location occurred. Further investigation revealed that in this study, regions were obtained based on SARS-CoV-2 GFF annotation provided by NCBI, with only 13,468–16,236 (ORF1B) inside NSP12 considered, leaving the rest 13,442–13,468 (ORF1A) not included; however, previous studies showed the shifting does not merely impact the exact location of amino acid mutations, rather only the perspective based on the ORF [17]. For instance, the P323L in NSP12 identified from previous and known studies was derived from full ORF1AB, whereas the P314L identified in this study was actually located in ORF1B only, as shown in Table 12 and Table 13.

A comparison of batch 3 samples running between Fast Pipeline and Normal Pipeline showed five amino acid mutations observed in three samples, represented as a yellow color in Table 14. First, samples S3-4, at position 386 inside the region of NSP4-ORF1AB, have a T/Y amino acid mutation using the Normal Pipeline and have a T amino acid mutation using the Fast Pipeline. Second, sample S3-9 has three different amino acid mutations. At position 1396 inside the region of NSP3-ORF1AB, samples S3-9, S1369L amino acid mutation is not detected using the Normal Pipeline but is detected using the Fast Pipeline. At position 43 inside the region of ORF8, sample S3-9, S43P amino acid mutation is detected using the Normal Pipeline, but not detected using the Fast Pipeline. At position 151 inside the nucleocapsid, P151S amino acid mutation is detected using the Normal Pipeline, but not detected using the Fast Pipeline. Third, in samples S3-14, at position 57 inside the ORF3A, Q57H amino acid mutation is not detected in the Normal Pipeline but detected using the Fast Pipeline.

Despite the occurrence of ambiguous amino acids, both pipelines work well and are capable of identifying specific mutations belonging to SARS-CoV-2. Prominent mutations with abundant studies during the COVID-19 pandemic, including P314L (NSP12-ORF1AB), D614G (spike glycoprotein), and Q57H (NS3-ORF3A), were found in all samples running both pipelines, as shown in Table 12, Table 13 and Table 14. D614G mutation in spike glycoprotein is the most studied among all, owing to its capabilities to enhance viral replications in epithelial cells, resulting in an increasing level of stability and enhancement of infectivity [18,19]. It is also well considered as the major circulating mutation in Indonesia [9]. Furthermore, D614G is also responsible for amino acid mutations in other regions as well, such as in P323L in RNA-dependent polymerase or NSP12, where it was associated with D614G as contributing factor in the viral infectivity [17,20]. On the other hand, Q57H mutation in NS3-ORF3A assists viruses in evading induction immune responses including interferon-stimulated gene, cytokine, and chemokine, as it causes truncation of ORF3B [21,22].

We present a comparison of fully complete runtime execution starting from quality control up until the end result of amino acids detection in both the Fast Pipeline and Normal Pipeline. As recorded in Table 15, Fast Pipeline, by default, successfully achieved the shortest time required to fully complete the pipeline from raw FASTQ data to the detection of amino acid mutations. All samples running the Fast Pipeline would require significantly less time than samples running the Normal Pipeline. Samples S3-14 were able to reach the shortest time with only 1 min to fully complete the pipelines.

## 5. Conclusions

This study evaluates the improved enrichment kit of Respiratory Virus Oligo Panel specified for Illumina NGS systems by directly detecting the SARS-CoV-2 genome inside clinical samples through the utilization of different bioinformatics pipelines called ‘Fast Pipeline’ and ‘Normal Pipeline’. We noted the advantages and drawbacks of each pipeline. Fast Pipeline ultimately works well in a situation where time is a critical factor. Its mesmerizing capabilities in shortening the time required to detect nucleotide substitutions and amino acid mutations are excellent, especially in tracing and detecting new SARS-CoV-2 variants. We discovered a higher number of reads mapped to the SARS-CoV-2 genome in the Fast Pipeline and merely as a contributing factor in a higher number of coverage depth and identified variations (SNPs, insertion, and deletion). Further study should be conducted concerning these underlying conditions and whether they might affect the results later on during downstream analysis, for instance, in the identification of high-quality SNPs. On the other hand, Normal Pipeline would require a longer time as it mapped reads to the human genome; however, it utilizes the standard metagenomics principles, filtering out reads of the human genome from samples to obtain pure viral genomes of SARS-CoV-2; therefore, the distribution of mapped reads are known and identified variations are accurate. Overall, both pipelines work well in the characterization of SARS-CoV-2 samples, including in the identification of major studied nucleotide substitutions and amino acid mutations. Furthermore, we noted a limitation to the unintegrated executable script; mainly, both bash and Python scripts in this study are separated entities from different environments. It is recommended in future studies to design a pipeline in an integrated framework, for instance, by using NextFlow, a workflow framework to combine all scripts into one fully integrated pipeline.

## Figures and Tables

**Figure 1 genes-13-01330-f001:**
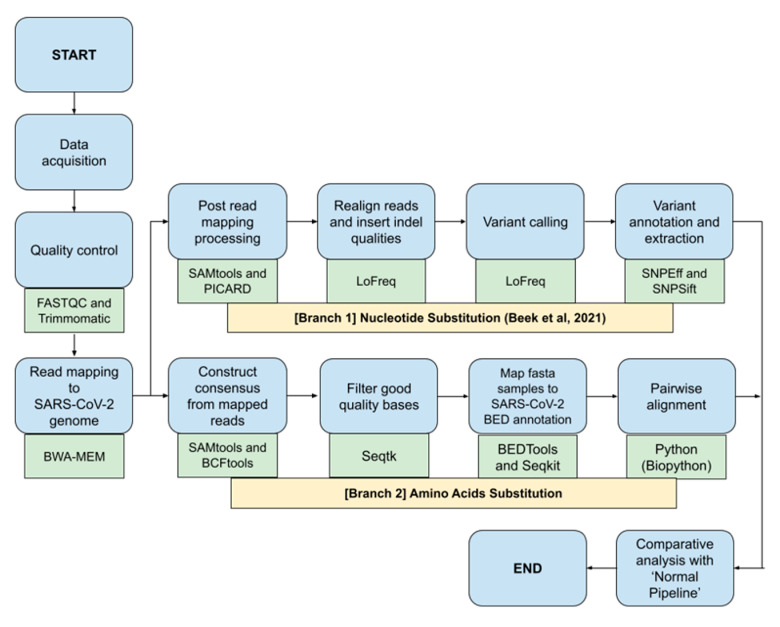
Fast Pipeline scheme; blue shapes represent the method; green shapes represent the tools used in each phase.

**Figure 2 genes-13-01330-f002:**
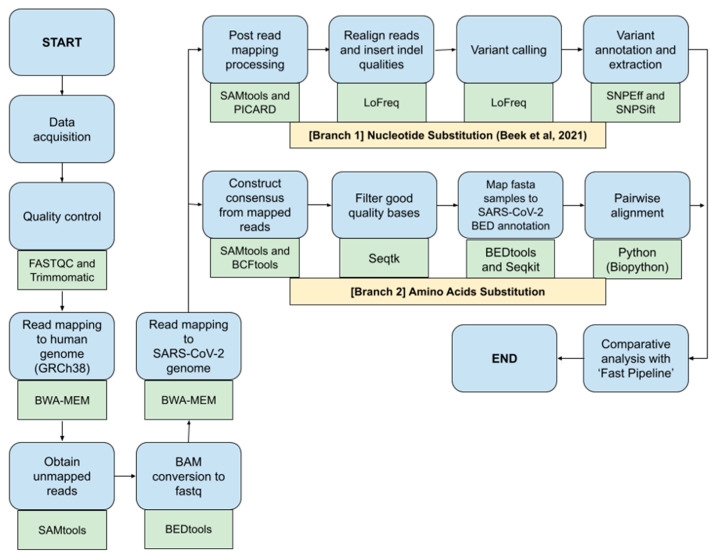
Normal Pipeline scheme; blue shapes represent the method; green shapes represent the tools used in each phase.

**Figure 3 genes-13-01330-f003:**
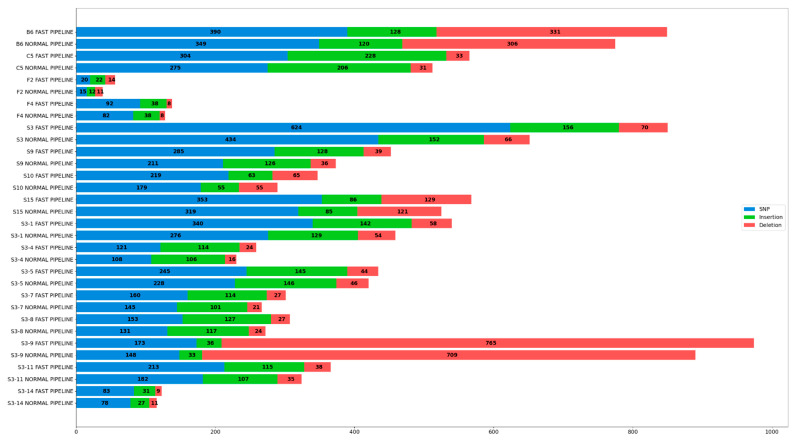
Variation (SNP, Insertion, and Deletion) Detected in All Samples Implemented in Both Pipelines.

**Figure 4 genes-13-01330-f004:**
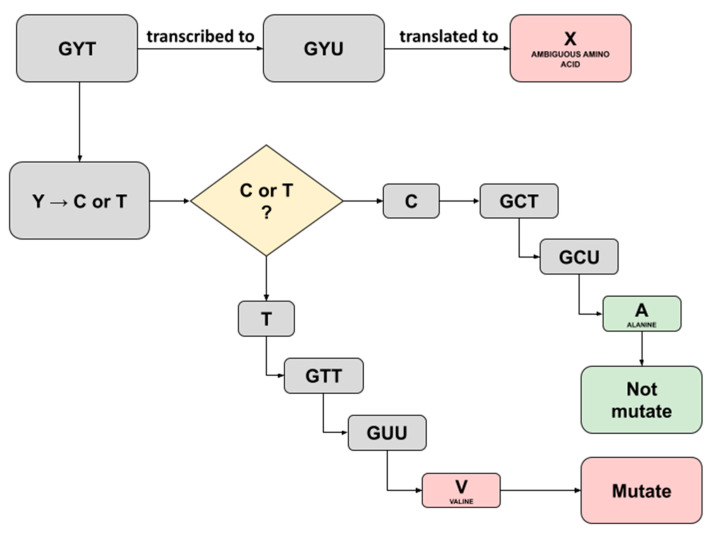
Illustration depicting the possible translation result of ambiguous amino acid X detected in Sample C5 running both Normal Pipeline and Fast Pipeline. An X amino acid was detected at position 54 region NS3-ORF3A.

**Figure 5 genes-13-01330-f005:**
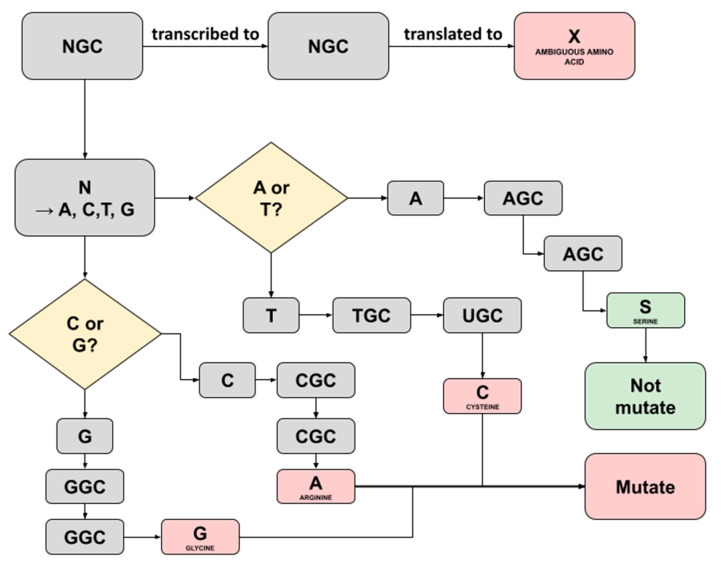
Illustration depicting the possible translation result of ambiguous amino acid X detected in Sample F2 running in Normal Pipeline. An X amino acid was detected at position 769 region NSP12-ORF1AB.

**Table 1 genes-13-01330-t001:** Data of patients with COVID-19 from Yogyakarta and Central Java that were involved in this study.

No	NGS Sample Code	NGS Batch	Sample ID	Sex	Age (Years)	Collection Date	Comorbid
1	B6	1	DIY-C25.2-02449	Male	77	22 June 2020	Yes
2	C5	1	DIY-C78.01481	Female	83	10 August 2020	Yes
3	F2	1	DIY-C25.2-00927	Male	30	16 May 2020	No
4	F4	1	KLN-C25.2-02538	Female	55	26 June 2020	Yes
5	S3	2	RSS-10001	Male	88	18 August 2020	Yes
6	S9	2	BBTKLPP-47964	Male	48	31 August 2020	Yes
7	S10	2	BBTKLPP-48651	Male	41	9 September 2020	No
8	S15	2	DIY-C78.00061	Female	49	16 June 2020	No
9	S3-1	3	DIY 1-58634	Male	65	18 September 2020	Yes
10	S3-4	3	DIY 1-24778	Male	34	23 December 2020	No
11	S3-5	3	DIY 1-10279	Male	77	7 September 2020	No
12	S3-7	3	DIY 1-10282	Female	42	7 September 2020	No
13	S3-8	3	DIY 1-24762	Female	48	23 December 2020	No
14	S3-9	3	RSS-10008	Male	58	27 December 2020	Yes
15	S3-11	3	DIY 1-24776	Female	34	23 December 2020	No
16	S3-14	3	53311	Female	81	9 September 2020	Yes

**Table 2 genes-13-01330-t002:** The overview of WGS data that were involved in the study.

NGS Sample Code	Batch	CT Value	Total Sequences (Paired-End Reads)	Sequence Length (bp)	% GC
B6	1	19.70	11,268,022	35–74	41
C5	1	16.90	2,707,228	35–74	42
F2	1	27.92	2,461,478	35–74	50
F4	1	24.68	1,366,538	35–74	45
S3	2	18.10	18,807,934	35–74	38
S9	2	19.64	7,827,098	35–74	46
S10	2	21.24	2,698,396	35–74	42
S15	2	22.31	6,111,408	35–74	46
S3-1	3	19.53	3,566,896	35–74	40
S3-4	3	13.27	1,167,562	35–74	38
S3-5	3	21.00	9,941,746	35–74	38
S3-7	3	21.55	1,669,316	35–74	39
S3-8	3	15.67	2,731,486	35–74	39
S3-9	3	22.27	4,748,810	35–74	45
S3-11	3	16.89	5,895,626	35–74	39
S3-14	3	17.73	376,514	35–74	44

**Table 3 genes-13-01330-t003:** Total sequences before and after quality control using Trimmomatic.

NGS Sample Code	Total Sequences		
	Before Trimming (Paired-End Reads)	Post-Trimming (QC) (Paired-End Reads)	Trimmed Sequence (%)
B6	11,268,022	11,184,784	0.74
C5	2,707,228	2,683,232	0.89
F2	2,461,478	2,440,518	0.85
F4	1,366,538	1,345,416	1.55
S3	18,807,934	18,387,180	2.24
S9	7,827,098	7,587,506	3.06
S10	2,698,396	2,590,256	4.01
S15	6,111,408	5,942,890	2.76
S3-1	3,566,896	3,502,824	1.80
S3-4	1,167,562	1,155,934	1.00
S3-5	9,941,746	9,807,834	1.35
S3-7	1,640,458	1,640,458	1.73
S3-8	2,731,486	2,696,200	1.29
S3-9	4,670,496	4,670,496	1.65
S3-11	5,816,070	5,816,070	1.35
S3-14	372,662	372,662	1.02
Average			1.70

**Table 4 genes-13-01330-t004:** The alignment statistics summary of unmapped and fully mapped reads in each sample’s post-read mapping to the SARS-CoV-2 genome (NC_045512.2).

NGS Sample Code	Unmapped to Sars-Cov-2 Genome		Fully Mapped to Sars-Cov-2 Genome	
	Number of Reads	Percentage (%)	Number of Reads	Percentage (%)
B6	2,028,393	18.14	9,156,391	81.86
C5	1,108,537	41.31	1,574,695	58.69
F2	2,391,210	97.98	49,308	2.02
F4	1,203,004	89.42	142,412	10.58
S3	548,965	2.99	17,838,215	97.01
S9	4,969,736	65.50	2,617,770	34.50
S10	1,452,990	56.09	1,137,266	43.91
S15	4,071,831	68.52	1,871,059	31.48
S3-1	830,959	23.72	2,671,865	76.28
S3-4	19,722	1.71	1,136,212	98.29
S3-5	205,582	2.10	9,602,252	97.90
S3-7	275,337	16.78	1,365,121	83.22
S3-8	175,137	6.50	2,521,063	93.50
S3-9	3,427,102	73.38	1,243,394	26.62
S3-11	576,452	9.91	5,239,618	90.09
S3-14	290,681	78.00	81,981	22.00

**Table 5 genes-13-01330-t005:** Distribution of reads in each sample post-read mapping to the human genome (GRCh38) and SARS-CoV-2 genome (NC_045512.2).

NGS Sample Code	Fully Mapped to Sars-Cov-2 Genome		Fully Mapped to Human Genome		Neither Both		Skipped during BAM to FASTQ Conversion	
	Number of Reads	Percentage (%)	Number of Reads	Percentage (%)	Number of Reads	Percentage (%)	Number of Reads	Percentage (%)
B6	8,743,980	78.18	2,435,133	21.77	3444	0.02	2227	0.02
C5	1,467,402	54.69	1,125,429	41.94	89,534	3.34	867	0.03
F2	38,668	1.58	2,399,180	98.31	2272	0.09	398	0.02
F4	134,956	10.03	1,210,132	89.94	108	0.01	220	0.02
S3	15,158,756	82.44	3,216,116	17.49	9700	0.05	2608	0.01
S9	2,363,094	31.14	5,214,314	68.72	7000	0.09	3098	0.04
S10	1,009,265	38.96	1,546,174	59.69	34,167	1.32	650	0.03
S15	1,676,134	28.20	4,235,721	71.27	28,482	0.48	2553	0.04
S3-1	2,321,562	66.28	1,180,022	33.69	502	0.01	738	0.02
S3-4	988,416	85.51	165,770	14.34	1706	0.15	42	0.00
S3-5	8,996,852	91.73	807,254	8.23	3020	0.03	708	0.01
S3-7	1,249,452	76.16	389,999	23.77	512	0.03	495	0.03
S3-8	2,332,361	86.51	345,223	12.80	18,489	0.69	127	0.00
S3-9	1,156,467	24.76	3,230,410	69.17	282,417	6.05	1202	0.03
S3-11	4,628,769	79.59	1,178,473	20.26	8509	0.15	319	0.01
S3-14	76,805	20.61	295,584	79.32	207	0.06	66	0.02
	Average	53.52	Average	45.67	Average	0.78	Average	0.02

**Table 6 genes-13-01330-t006:** Read mapping coverage results.

NGS Sample Code	Read Mapping Coverage (Times)		Difference Fast vs. Normal (%)
	Fast Pipeline	Normal Pipeline	
B6	22,352.5	21,357.2	4.70
C5	3833.9	3576.5	7.20
F2	115	94.6	21.6
F4	347.7	329.8	5.40
S3	43,244.4	36,843	17.4
S9	6350.02	5744.18	10.5
S10	2756.31	2457.99	12.1
S15	4545.72	4077.32	11.5
S3-1	6494.44	5653.07	14.9
S3-4	2764.01	2410.1	14.7
S3-5	23,481.7	22,016.8	6.60
S3-7	3333.82	3054.74	9.10
S3-8	6163.56	5706.29	8.00
S3-9	3033.52	2824.06	7.40
S3-11	12,794.7	11,323.3	13.00
S3-14	199.371	186.96	6.60
Average			10.66

**Table 7 genes-13-01330-t007:** Statistics Summary of Variation (SNP, Insertion, and Deletion) Detected in All Samples Implemented in Both Pipelines.

NGS Sample Code	Fast Pipeline				Normal Pipeline				Difference Fast vs. Normal (%)			
	#SNP	#Insertion	#Deletion	All Variation	#SNP	#Insertion	#Deletion	All Variation	#SNP	#Insertion	#Deletion	All Variation
B6	390	128	331	849	349	120	306	775	11.75	6.67	8.17	9.55
C5	304	228	33	565	275	206	31	512	10.55	10.68	6.45	10.35
F2	20	22	14	56	15	12	11	38	33.33	83.33	27.27	47.37
F4	92	38	8	138	82	38	8	128	12.20	0.00	0.00	7.81
S3	624	156	70	850	434	152	66	652	43.78	2.63	6.06	30.37
S9	285	128	39	452	211	126	36	373	35.07	1.59	8.33	21.18
S10	219	63	65	347	179	55	55	289	22.35	14.55	18.18	20.07
S15	353	86	129	568	319	85	121	525	10.66	1.18	6.61	8.19
S3-1	340	142	58	540	276	129	54	459	23.19	10.08	7.41	17.65
S3-4	121	114	24	259	108	106	16	230	12.04	7.55	50.00	12.61
S3-5	245	145	44	434	228	146	46	420	7.46	−0.68	−4.35	3.33
S3-7	160	114	27	301	145	101	21	267	10.34	12.87	28.57	12.73
S3-8	153	127	27	307	131	117	24	272	16.79	8.55	12.50	12.87
S3-9	173	36	765	974	148	33	709	890	16.89	9.09	7.90	9.44
S3-11	213	115	38	366	182	107	35	324	17.03	7.48	8.57	12.96
S3-14	83	31	9	123	78	27	11	116	6.41	14.81	−18.18	6.03
Average									18.11	11.90	10.84	15.16

**Table 8 genes-13-01330-t008:** Identified SNPs in All Batch 1 Samples Running All Pipelines.

POSITION	5’UTR	NSP3-ORF1AB				NSP5-ORF1AB		NSP12-ORF1AB					NSP13-ORF1AB	NSP14-ORF1AB		SPIKE-S	NS3-ORF3A			MATRIX-M		NS7A-ORF7A	NP-N		
POSITION	241	3037	3529	4754	5184	10201	10507	14055	14292	14408	14694	15406	17964	18744	18877	23403	25553	25563	25687	26735	26867	27610	28735	28752	29209
REFERENCE (NC_045512.2)	C	C	T	C	C	G	C	G	C	C	C	G	G	C	C	A	C	G	G	C	A	C	T	A	A
B6 FAST PIPELINE	**T**	**T**	T	C	**T**	G	**T**	G	C	**T**	C	G	G	**T**	**T**	**G**	C	**T**	G	**T**	**G**	C	T	A	A
B6 NORMAL PIPELINE	**T**	**T**	T	C	**T**	G	**T**	G	C	**T**	C	G	G	**T**	**T**	**G**	C	**T**	G	**T**	**G**	C	T	A	A
C5 FAST PIPELINE	**T**	**T**	**C**	**T**	C	G	C	G	**T**	**T**	**T**	**T**	**T**	C	**T**	**G**	**T**	**T**	**T**	**T**	A	C	**C**	**G**	A
C5 NORMAL PIPELINE	**T**	**T**	**C**	**T**	C	G	C	G	**T**	**T**	**T**	**T**	**T**	C	**T**	**G**	**T**	**T**	**T**	**T**	A	C	**C**	**G**	A
F2 FAST PIPELINE	C	C	T	C	C	**T**	C	G	C	C	C	G	G	C	C	A	C	G	G	C	A	C	T	A	**G**
F2 NORMAL PIPELINE	C	C	T	C	C	**T**	C	G	C	C	C	G	G	C	C	A	C	G	G	C	A	C	T	A	**G**
F4 FAST PIPELINE	**T**	**T**	T	C	**T**	G	**T**	**T**	C	**T**	C	G	G	**T**	**T**	**G**	C	**T**	G	**T**	**G**	**T**	T	A	A
F4 NORMAL PIPELINE	**T**	**T**	T	C	**T**	G	**T**	**T**	C	**T**	C	G	G	**T**	**T**	**G**	C	**T**	G	**T**	**G**	**T**	T	A	A

**Table 9 genes-13-01330-t009:** (a) Identified SNPs in All Batch 2 Samples Running All Pipelines (Part 1). (b) Identified SNPs in All Batch 2 Samples Running All Pipelines (Part 2).

(a) Identified SNPs in All Batch 2 Samples Running All Pipelines (Part 1)																										
REGION	5’UTR	NSP1-ORF1AB			NSP3-ORF1AB						NSP5-ORF1AB		NSP6-ORF1AB	NSP8-ORF1AB		NSP9-ORF1AB	NSP12-ORF1AB							NSP13-ORF1AB		
POSITION	241	1545	2263	2512	3037	4084	5184	5784	6312	7639	10089	10507	11083	12152	12439	12809	13730	14120	14183	14408	15543	15765	16156	16395	16647	16694
REFERENCE (NC_045512.2)	C	C	C	A	C	C	C	C	C	C	A	C	G	G	C	C	C	C	C	C	G	A	A	A	G	C
S3 FAST PIPELINE	**T**	**T**	C	A	**T**	**T**	**T**	C	C	C	**G**	**T**	G	G	C	C	C	C	**T**	**T**	G	A	A	**T**	**T**	C
S3 NORMAL PIPELINE	**T**	**T**	C	A	**T**	**T**	**T**	C	C	C	**G**	**T**	G	G	C	C	C	C	**T**	**T**	G	A	A	**T**	**T**	C
S9 FAST PIPELINE	C	C	C	**G**	C	C	C	C	**A**	C	A	C	**T**	**A**	**T**	**T**	**T**	C	C	C	G	A	G	A	G	**T**
S9 NORMAL PIPELINE	C	C	C	**G**	C	C	C	C	**A**	C	A	C	**T**	**A**	**T**	**T**	**T**	C	C	C	G	A	**G**	A	G	**T**
S10 FAST PIPELINE	**T**	C	C	A	**T**	C	C	C	C	C	A	C	G	G	C	C	C	**T**	C	**T**	G	**G**	A	A	G	C
S10 NORMAL PIPELINE	**T**	C	C	A	**T**	C	C	C	C	C	A	C	G	G	C	C	C	**T**	C	**T**	G	**G**	A	A	G	C
S15 FAST PIPELINE	**T**	C	**T**	A	**T**	C	**T**	**T**	C	**T**	A	**T**	G	G	C	C	C	C	C	**T**	**T**	A	A	A	G	C
S15 NORMAL PIPELINE	**T**	C	**T**	A	**T**	C	**T**	**T**	C	**T**	A	**T**	G	G	C	C	C	C	C	**T**	**T**	A	A	A	G	C
**(b) Identified SNPs in All Batch 2 Samples Running All Pipelines (Part 2)**																										
**REGION**	**NSP14-ORF1AB**				**NSP15-A1-ORF1AB**		**SPIKE-S**									**NS3-ORF3A**		**MATRIX-M**		**ORF8**	**NP-N**					**ORF10**
POSITION	18744	18877	19002		20124		21652	21742	21748	21809	22200	22334	23403	23593	23929	25563	26056	26735	26867	28073	28311	28628	28851	28975		29642
REFERENCE (NC_045512.2)	C	C	A		T	T		C	T	G	T	T	A	G	C	G	G	C	A	G	C	G	G	G		C
S3 FAST PIPELINE	**T**	**T**	A		T	T		**T**	T	G	**C**	T	**G**	G	C	**T**	G	**T**	**G**	G	C	**T**	**T**	G		C
S3 NORMAL PIPELINE	**T**	**T**	A		T	T		**T**	T	G	**C**	T	**G**	G	C	**T**	G	**T**	G	G	C	**T**	**T**	G		C
S9 FAST PIPELINE	C	C	G		**C**	C		C	**C**	G	T	**C**	**G**	G	**T**	G	G	C	A	**A**	**T**	G	G	G		C
S9 NORMAL PIPELINE	C	C	**G**		**C**	**C**		C	**C**	G	T	**C**	**G**	G	**T**	G	G	C	A	**A**	**T**	G	G	G		C
S10 FAST PIPELINE	C	**T**	A		T	T		C	T	**C**	T	T	**G**	**T**	C	**T**	**T**	**T**	A	G	C	G	G	**T**		**T**
S10 NORMAL PIPELINE	C	**T**	A		T	T		C	T	C	T	T	**G**	**T**	C	**T**	**T**	**T**	A	G	C	G	G	**T**		**T**
S15 FAST PIPELINE	**T**	**T**	A		T	T		C	T	G	T	T	**G**	G	C	**T**	G	**T**	A	G	C	G	G	G		C
S15 NORMAL PIPELINE	**T**	**T**	A		T	T		C	T	G	T	T	**G**	G	C	**T**	G	**T**	A	G	C	G	G	G		C

**Table 10 genes-13-01330-t010:** Identified SNPs in All Batch 3 Samples Running All Pipelines.

REGION	NSP3-ORF1AB					NSP4-ORF1AB			NSP5A-ORF1AB		NSP6-ORF1AB		NSP7-ORF1AB	NSP12-ORF1AB				NSP15-ORF1AB				SPIKE GLYCOPROTEIN								ORF3A			ORF8	NP-N							
POSITION	3305	5184	5554	6309	6906	9701	9710	9711	10313	10904	10995	11219	11991	14120	14408	14741	15848	19793	19794	20443	20611	21575	22200	23042	23270	23403	23599	23629	25337	25563	25590	25904	28020	28628	28655	28724	28851	28881	28883	28975	28977
REFERENCE (NC_045512.2)	A	C	G	G	C	A	T	C	C	A	A	A	A	C	C	C	C	G	G	G	C	C	T	T	G	A	T	T	G	G	A	C	T	G	G	C	G	G	G	G	C
S3-1 FAST PIPELINE	A	**T**	G	G	C	A	T	C	C	A	A	A	A	C	**T**	**T**	C	G	G	**T**	**T**	C	**C**	T	G	**G**	T	T	G	**T**	A	C	T	**T**	G	C	**T**	G	G	G	C
S3-1 NORMAL PIPELINE	A	**T**	G	G	C	A	T	C	C	A	A	A	A	C	**T**	**T**	C	G	G	**T**	**T**	C	**C**	T	G	**G**	T	T	G	**T**	A	C	T	**T**	G	C	**T**	G	G	G	C
S3-4 FAST PIPELINE	A	C	G	G	C	A	**A**	**A**	**T**	A	A	A	A	**T**	T	C	**T**	G	G	G	C	C	T	T	**T**	**G**	**A**	T	**T**	**T**	A	C	T	G	G	C	G	G	G	G	**T**
S3-4 NORMAL PIPELINE	A	C	G	G	C	A	**A**	**A**	**T**	A	A	A	A	**T**	T	C	**T**	G	G	G	C	C	T	T	**T**	**G**	**A**	T	**T**	**T**	A	C	T	G	G	C	G	G	G	G	**T**
S3-5 FAST PIPELINE	**C**	C	G	G	C	**G**	T	C	C	A	A	A	A	**T**	**T**	C	**T**	G	G	G	C	**T**	T	T	G	**G**	T	T	G	**T**	**T**	**T**	T	G	**T**	C	G	G	G	G	**T**
S3-5 NORMAL PIPELINE	**C**	C	G	G	C	**G**	T	C	C	A	A	A	A	**T**	**T**	C	**T**	G	G	G	C	**T**	T	T	G	**G**	T	T	G	**T**	**T**	**T**	T	G	**T**	C	G	G	G	G	**T**
S3-7 FAST PIPELINE	**C**	C	G	G	C	**G**	T	C	C	A	A	A	A	**T**	**T**	C	**T**	G	G	G	C	**T**	T	T	G	**G**	T	T	G	**T**	**T**	**T**	T	G	**T**	C	G	G	G	G	**T**
S3-7 NORMAL PIPELINE	**C**	C	G	G	C	**G**	T	C	C	A	A	A	A	**T**	**T**	C	**T**	G	G	G	C	**T**	T	T	G	**G**	T	T	G	**T**	**T**	**T**	T	G	**T**	C	G	G	G	G	**T**
S3-8 FAST PIPELINE	A	**T**	G	**C**	C	A	T	C	C	A	A	**G**	A	C	**T**	C	C	G	G	G	C	C	T	**C**	G	**G**	T	**G**	G	**T**	A	C	T	**T**	G	C	G	G	G	**T**	C
S3-8 NORMAL PIPELINE	A	**T**	G	**C**	C	A	T	C	C	A	A	**G**	A	C	**T**	C	C	G	G	G	C	C	T	**C**	G	**G**	T	**G**	G	**T**	A	C	T	**T**	G	C	G	G	G	**T**	C
S3-9 FAST PIPELINE	A	C	**T**	G	**T**	A	T	C	C	**G**	**G**	A	**G**	C	**T**	C	C	**T**	**T**	G	C	C	T	T	G	**G**	T	T	G	G	A	C	**T**	G	G	**C**	G	**A**	**C**	G	C
S3-9 NORMAL PIPELINE	A	C	**T**	G	**C**	A	T	C	C	**G**	**G**	A	**G**	C	**T**	C	C	**T**	**T**	G	C	C	T	T	G	**G**	T	T	G	G	A	C	**C**	G	G	**T**	G	**A**	**C**	G	C
S3-11 FAST PIPELINE	A	C	G	G	C	A	**A**	**A**	**T**	A	A	A	A	**T**	**T**	C	**T**	G	G	G	C	C	T	T	**T**	**G**	**A**	T	**T**	**T**	A	C	T	G	G	C	G	G	G	G	**T**
S3-11 NORMAL PIPELINE	A	C	G	G	C	A	**A**	**A**	**T**	A	A	A	A	**T**	**T**	C	**T**	G	G	G	C	C	T	T	**T**	**G**	**A**	T	**T**	**T**	A	C	T	G	G	C	G	G	G	G	**T**
S3-14 FAST PIPELINE	A	**T**	G	G	C	A	T	C	C	A	A	A	A	C	C	C	C	G	G	G	C	**T**	T	T	G	**G**	T	T	G	**T**	A	C	T	**T**	G	C	**T**	G	G	G	C
S3-14 NORMAL PIPELINE	A	**T**	G	G	C	A	T	C	C	A	A	A	A	C	C	C	C	G	G	G	C	**T**	T	T	G	**G**	T	T	G	**G**	A	C	T	**T**	G	C	**T**	G	G	G	C

**Table 11 genes-13-01330-t011:** Result of consensus sequences constructed by using a combination of SAMtools and BEDtools.

NGS Sample Code	Length of Consensus Sequence (bp)	
	Fast Pipeline	Normal Pipeline
B6	29,903	29,894
C5	29,903	29,892
F2	29,903	29,853
F4	29,903	29,877
S3	29,903	29,890
S9	29,903	29,892
S10	29,903	29,870
S15	29,903	29,879
S3-1	29,903	29,892
S3-4	29,903	29,870
S3-5	29,903	29,877
S3-7	29,903	29,867
S3-8	29,903	29,870
S3-9	29,903	29,892
S3-11	29,903	29,870
S3-14	29,903	29,869

**Table 12 genes-13-01330-t012:** Identified Amino Acids Mutations in Batch 1 Samples Running All Pipelines.

REGION	5’UTR	NSP3-ORF1AB		NSP5-ORF1AB	NSP12-ORF1AB			NSP13-ORF1AB	SPIKE-S	NS3-ORF3A			NS7A-ORF7A	NP-N
POSITION	81	679	822	49	314	646	769	576	614	54	57	99	73	160
REFERENCE (NC_045512.2)	R	P	P	M	P	A	S	M	D	A	Q	A	H	Q
B6 FAST PIPELINE	**C**	P	**L**	M	**L**	A	S	M	**G**	A	**H**	A	H	Q
B6 NORMAL PIPELINE	**C**	P	**L**	M	**L**	A	S	M	**G**	A	**H**	A	H	Q
C5 FAST PIPELINE	**C**	**S**	P	M	**L**	**S**	S	**I**	**G**	**X**	**H**	**S**	H	**R**
C5 NORMAL PIPELINE	**C**	**S**	P	M	**L**	**S**	S	**I**	**G**	**X**	**H**	**S**	H	**R**
F2 FAST PIPELINE	R	P	P	**I**	P	A	**S**	M	D	A	Q	A	H	Q
F2 NORMAL PIPELINE	R	P	P	**I**	P	A	**X**	M	D	A	Q	A	H	Q
F4 FAST PIPELINE	R	P	**L**	M	**L**	A	S	M	**G**	A	**H**	A	**Y**	Q
F4 NORMAL PIPELINE	R	P	**L**	M	**L**	A	S	M	**G**	A	**H**	A	**Y**	Q

**Table 13 genes-13-01330-t013:** Identified Amino Acids Mutations in Batch 2 Samples Running All Pipelines.

REGION	5’UTR	NSP3-ORF1AB			NSP5-ORF1AB	NSP6	NSP8-ORF1AB	NSP9	NSP12-ORF1AB					NSP13-ORF1AB	SPIKE-S					NS3-ORF3A		NP-N				ORF8
POSITION	81	822	1022	1198	12	37	21	42	88	218	239	314	897	153	83	213	258	614	677	57	222	13	119	193	234	29
REFERENCE (NC_045512.2)	R	P	T	T	K	L	A	L	A	P	T	P	M	T	V	V	W	D	Q	Q	D	P	A	S	M	Q
S3 FAST PIPELINE	**C**	**L**	T	T	**R**	L	A	L	A	P	**I**	**L**	M	T	V	**A**	W	**G**	Q	**H**	D	P	**S**	**I**	M	Q
S3 NORMAL PIPELINE	**C**	**L**	T	T	**R**	L	A	L	A	P	**I**	**L**	M	T	V	**A**	W	**G**	Q	**H**	D	P	**S**	**I**	M	Q
S9 FAST PIPELINE	R	P	T	**K**	K	**F**	**T**	**F**	**V**	P	T	P	**V**	**I**	V	V	**R**	**G**	Q	Q	D	**L**	A	S	M	Q
S9 NORMAL PIPELINE	R	P	T	**K**	K	**F**	**T**	**F**	**V**	P	T	P	**V**	**I**	V	V	**R**	**G**	Q	Q	D	**L**	A	S	M	Q
S10 FAST PIPELINE	**C**	P	T	T	K	L	A	L	A	**L**	T	**L**	M	T	**L**	V	W	**G**	**H**	**H**	**Y**	P	A	S	**I**	*****
S10 NORMAL PIPELINE	**C**	P	T	T	K	L	A	L	A	**L**	T	**L**	M	T	**L**	V	W	**G**	**H**	**H**	**Y**	P	A	S	**I**	*****
S15 FAST PIPELINE	**C**	**L**	**I**	T	K	L	A	L	A	P	T	**L**	M	T	V	V	W	**G**	Q	**H**	D	P	A	S	M	Q
S15 NORMAL PIPELINE	**C**	**L**	**I**	T	K	L	A	L	A	P	T	**L**	M	T	V	V	W	**G**	Q	**H**	D	P	A	S	M	Q

**Table 14 genes-13-01330-t014:** Identified Amino Acids Mutations in Batch 3 Samples Running All Pipelines.

REGION	NSP3-ORF1AB					NSP4-ORF1AB		NSP5A-ORF1AB		NSP6-ORF1AB		NSP7-ORF1AB	NSP12-ORF1AB				NSP15-ORF1AB			SPIKE GLYCOPROTEIN								ORF3A			ORF8	NP-N							
POSITION	196	822	945	1197	1369	383	386	87	284	8	83	50	227	323	434	803	58	275	331	5	213	494	570	614	679	689	1259	57	66	171	43	119	128	151	193	203	204	234	235
REFERENCE (NC_045512.2)	M	P	K	S	S	I	S	L	S	K	M	E	P	P	S	T	W	V	L	L	V	S	A	D	N	S	D	Q	K	S	S	A	D	P	S	R	G	M	S
S3-1 FAST PIPELINE	M	**L**	K	S	S	I	S	L	S	K	M	E	P	**L**	**F**	T	W	**F**	**F**	L	**A**	S	A	**G**	N	S	D	**H**	K	S	S	**S**	D	P	**I**	R	G	M	S
S3-1 NORMAL PIPELINE	M	**L**	K	S	S	I	S	L	S	K	M	E	P	**L**	**F**	T	W	**F**	**F**	L	**A**	S	A	**G**	N	S	D	**H**	K	S	S	**S**	D	P	**I**	R	G	M	S
S3-4 FAST PIPELINE	M	P	K	S	S	I	**T**	**F**	S	K	M	E	**L**	**L**	S	**I**	W	V	L	L	V	S	**S**	**G**	**K**	S	**Y**	**H**	**K**	S	S	A	D	P	S	R	G	M	**F**
S3-4 NORMAL PIPELINE	M	P	K	S	S	I	**T/Y**	**F**	S	K	M	E	**L**	**L**	S	**I**	W	V	L	L	V	S	**S**	**G**	**K**	S	**Y**	**H**	**K**	S	S	A	D	P	S	R	G	M	**F**
S3-5 FAST PIPELINE	**L**	P	K	S	S	**V**	S	L	S	K	M	E	**L**	**L**	S	**I**	W	V	L	**F**	V	S	A	**G**	N	S	D	**H**	**N**	**L**	S	A	**Y**	P	S	R	G	M	**F**
S3-5 NORMAL PIPELINE	**L**	P	K	S	S	**V**	S	L	S	K	M	E	**L**	**L**	S	**I**	W	V	L	**F**	V	S	A	**G**	N	S	D	**H**	**N**	**L**	S	A	**Y**	P	S	R	G	M	**F**
S3-7 FAST PIPELINE	**L**	P	K	S	S	**V**	S	L	S	K	M	E	**L**	**L**	S	**I**	W	V	L	**F**	V	S	A	**G**	N	S	D	**H**	**N**	**L**	S	A	**Y**	P	S	R	G	M	**F**
S3-7 NORMAL PIPELINE	**L**	P	K	S	S	**V**	S	L	S	K	M	E	**L**	**L**	S	**I**	W	V	L	**F**	V	S	A	**G**	N	S	D	**H**	**N**	**L**	S	A	**Y**	P	S	R	G	M	**F**
S3-8 FAST PIPELINE	M	**L**	K	**T**	S	I	S	L	S	K	**V**	E	P	**L**	S	T	W	V	L	L	V	**P**	A	**G**	N	**R**	D	**H**	K	S	S	**S**	D	P	S	R	G	**I**	S
S3-8 NORMAL PIPELINE	M	**L**	K	**T**	S	I	S	L	S	K	**V**	E	P	**L**	S	T	W	V	L	L	V	**P**	A	**G**	N	**R**	D	**H**	K	S	S	**S**	D	P	S	R	G	**I**	S
S3-9 FAST PIPELINE	M	P	**N**	S	**L**	I	S	L	**G**	**R**	M	**G**	P	**L**	S	T	**L/C**	V	L	L	V	S	A	**G**	N	S	D	Q	K	S	**S**	A	D	**P**	S	**K**	**R**	M	S
S3-9 NORMAL PIPELINE	M	P	**N**	S	**S**	I	S	L	**G**	**R**	M	**G**	P	**L**	S	T	**L/C**	V	L	L	V	S	A	**G**	N	S	D	Q	K	S	**P**	A	D	**S**	S	**K**	**R**	M	S
S3-11 FAST PIPELINE	M	P	K	S	S	I	**T/Y**	**F**	S	K	M	E	**L**	**L**	S	**I**	W	V	L	L	V	S	**S**	**G**	**K**	S	**Y**	**H**	K	S	S	A	D	P	S	R	G	M	**F**
S3-11 NORMAL PIPELINE	M	P	K	S	S	I	**T/Y**	**F**	S	K	M	E	**L**	**L**	S	**I**	W	V	L	L	V	S	**S**	**G**	**K**	S	**Y**	**H**	K	S	S	A	D	P	S	R	G	M	**F**
S3-14 FAST PIPELINE	M	**L**	K	S	S	I	S	L	S	K	M	E	P	P	S	T	W	V	L	**F**	V	S	A	**G**	N	S	D	**H**	K	S	S	**S**	D	P	**I**	R	G	M	S
S3-14 NORMAL PIPELINE	M	**L**	K	S	S	I	S	L	S	K	M	E	P	P	S	T	W	V	L	**F**	V	S	A	**G**	N	S	D	**Q**	K	S	S	**S**	D	P	**I**	R	G	M	S

**Table 15 genes-13-01330-t015:** Total time required to fully complete each pipeline in detecting nucleotide substitution and amino acids mutation.

NGS Sample Code	Running Time (s)	
	Fast Pipeline	Normal Pipeline
B6	1778.0	5991.3
C5	574.3	3980.5
F2	324.5	3924.1
F4	286.4	3539.5
S3	3060.1	6521.0
S9	1036.8	4755.5
S10	537.5	3747.3
S15	848.9	4356.8
S3-1	552.2	4864.7
S3-4	256.3	4461.3
S3-5	1427.8	6377.2
S3-7	330.6	4416.8
S3-8	489.6	4824.5
S3-9	486.5	4752.9
S3-11	879.6	5394.5
S3-14	57.6	4190.2

## Data Availability

The sequence and metadata are shared through GISAID (www.gisaid.org, accessed on 1 March 2021).

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
