# Peer review of "A Comparison of Bioinformatics Pipelines for Enrichment Illumina Next Generation Sequencing Systems in Detecting SARS-CoV-2 Virus Strains"

_genes, 2022, doi:10.3390/genes13081330_

Round 1
Reviewer 1 Report
The draft " A Comparison of Bioinformatics Pipelines for Enrichment Illumina Next Generation Sequencing Systems in Detecting SARS-CoV-2 Virus Strains " written by Afiahayatiand colleagues focused on the potential Next generation Sequencing Systems in Detecting SARS-CoV-2 Virus Strains. They used many different bioinformatics pipelines named as ‘Fast Pipeline’ and ‘Normal Pipeline’ to SARS-CoV-2. Several modifications are needed before this draft being published on genes:
1. Improve abstract part as it does not represent the study results and conclusion. 2. A total of 8 patients with COVID-19 were involved in this study range between the age of 30 to 88. Number of sample might be increased to have better results. Secondly age limit is huge, secondly there might be comorbitities which is not mentioned. 3. How CT Value were calculated. Add this in supplementary file and mentioned the standard genes used in RT-PCR. 4. Report updated citation in introduction and results.Author Response
Please see the attachment.

Reviewer 2 Report
Manuscript "A Comparison of Bioinformatics Pipelines for Enrichment Illumina Next Generation Sequencing Systems in Detecting SARS-CoV-2 Virus Strains" is very interesting and importing.
General comments:
Authors compared different bioinformatics pipelines towards target enrichment workflow by Illumina. The NGS data were obtained from 8 hospitalized patients in Yogyakarta and Central Java, tested positive for SARS-CoV-2 and took Real Time - Polymerase Chain Reaction (RT-PCR) swab test between May - September 2020. The first pipeline dubbed as 'Fast Pipeline' will directly map the raw NGS data to the SARS-CoV-2 reference genome. While the second 'Normal Pipeline' will map the raw NGS data to the human genome and proceed to map subsequent unmapped reads to the SARS-CoV-2 reference genome. The comparison in-between pipelines observed all the way to identification of nucleotide substitutions and amino acids mutations.
Detailed comments:
Quality of Figure 1 is poor.
Quality of Figure 2 is poor.
Tables 4 and 5: Explain "x".
Table 8: Explain "nt".
Unfortunately, in paper is lack of statistical analysis.
Paper needs major revision.
Round 2
Reviewer 2 Report
Now, all is ok.